# Penetration Efficiency of Antitumor Agents in Ovarian Cancer Spheroids: The Case of Recombinant Targeted Toxin DARPin-LoPE and the Chemotherapy Drug, Doxorubicin

**DOI:** 10.3390/pharmaceutics11050219

**Published:** 2019-05-07

**Authors:** Evgeniya Sokolova, Olga Kutova, Alena Grishina, Anton Pospelov, Evgeniy Guryev, Alexey Schulga, Sergey Deyev, Irina Balalaeva

**Affiliations:** 1Institute of Biology and Biomedicine, Lobachevsky State University of Nizhny Novgorod, 23 Gagarin ave., Nizhny Novgorod 603950, Russia; kutovaom@gmail.com (O.K.); puzhikhina.alena@gmail.com (A.G.); eso103163@gmail.com (A.P.); ksonm06@yandex.ru (E.G.); irin-b@mail.ru (I.B.); 2Laboratory of molecular immunology, Institute of Bioorganic Chemistry of the Russian Academy of Sciences, 16/10 Miklukho-Maklay St., Moscow 117997, Russia; schulga@gmail.com (A.S.); deyev@mail.ibch.ru (S.D.); 3Institute of Molecular Medicine, I.M. Sechenov First Moscow State Medical University, 8-2 Trubetskaya str., Moscow 119991, Russia; 4Research Nuclear Reactor Center, National Research Tomsk Polytechnic University, 30 Lenin ave., Tomsk 634050, Russia; 5Institute of Engineering Physics for Biomedicine (PhysBio), National Research Nuclear University “MEPhI”, 31 Kashirskoe shosse, Moscow 115409, Russia

**Keywords:** drug penetration into tumor, HER2, spheroid, targeted therapy, targeted toxin, DARPin, *Pseudomonas* exotoxin A

## Abstract

The efficiency of delivering a therapeutic agent into a tumor is among the crucial factors determining the prospects for its clinical use. This problem is particularly acute in the case of targeted antitumor agents since many of them are high-molecular-weight compounds. In this work, the penetration of therapeutic agents of two distinct molecular weights into the spheroids of ovarian adenocarcinoma overexpressing human epidermal growth factor receptor 2 (HER2) was studied. It was shown that the low-molecular-weight chemotherapy drug, doxorubicin (~0.5 kDa), effectively penetrates through almost the entire depth of a 300 to 400 μm spheroid, while the penetration depth of the HER2-specific recombinant targeted toxin, DARPin-LoPE (~42 kDa), is only a few surface layers of cells and does not exceed 70 μm. The low penetration of the targeted toxin into spheroid was shown along with a significant decrease in its efficiency against the three-dimensional tumor spheroid as compared with the two-dimensional monolayer culture. The approaches to increasing the accumulation of agents in the tumor are presented and prospects of their use in order to improve the effectiveness of therapy are discussed.

## 1. Introduction

Targeted therapy is an actively developing approach to the treatment of malignant neoplasms based on the use of therapeutic agents, capable of selective eliminating tumor cells in the body. A wide number of agents are proposed for targeted antitumor therapy, showing high efficacy in vitro. However, when moving to in vivo studies and further to clinical practice, a significant decrease in their effectiveness is revealed [1]. Such a decrease in efficiency may be caused by a number of factors associated with the structure and biology of the tumor: The presence of numerous contacts of tumor cells with each other, with stromal cells, and with components of the extracellular matrix; the presence of foci of hypoxia and acidosis; an inhomogeneous distribution of gases and nutrients in the tumor mass; and the lack of adequate metabolite outflow. As a result, malignant cells with different microenvironment within the tumor node develop changes in genetic and metabolic profiles. The heterogeneity of cell populations, in turn, is one of the key factors of the tumor resistance to treatment. Tumor tissues are also characterized by a high interstitial fluid pressure caused from the aberrant vascular bed, which leads to the hindered diffusion and rapid elimination of the antitumor agent from the tumor mass [2].

One of the most studied and clinically important target molecules is human epidermal growth factor receptor 2 (HER2), which belongs to the epidermal growth factor receptor (EGFR) family. HER2 is overexpressed by many types of carcinomas and associated with poor prognosis and low patient survival [3,4]. Agents for HER2-targeted therapy are presented by low-molecular-weight compounds (tyrosine kinase inhibitors) and high-molecular-weight compounds (monoclonal antibodies and targeted toxins based on natural toxic proteins). For large molecules, a three-dimensional tumor structure determines, first, the physical limitations of penetration within the tumor and interactions with cells. The presence of a large number of tight cell–cell junctions causes, on the one hand, a reduction in the distance between cells, and on the other, blocking of the surface molecular targets to which these agents have been addressed [5]. One of the most malignant HER2-overexpressing tumors is ovarian carcinoma. Due to peculiar features of female physiology, it is very difficult to diagnose this type of cancer in early stages as no symptoms are manifested, so commonly the disease is diagnosed at a late stage. Global statistics shows less than a 45% survival of women suffering from ovarian carcinoma in a 5-year period [6].

This paper is devoted to the study of the penetration of therapeutic agents of two distinct molecular weights into the tumor spheroids of HER2-overexpressing human ovarian carcinoma. The tumor penetration depth of the high-molecular-weight targeted toxin, DARPin-LoPE (~42 kDa), reached 50 to 80 μm after 24 h of incubation, which is significantly poorer than the penetration of the low-molecular-weight chemotherapeutic drug, doxorubicin (~0.5 kDa). We suppose that the modest penetration depth of the agent into the spheroid might cause a significant decrease in its efficiency when transiting from 2D to 3D in vitro tumor models.

## 2. Materials and Methods

### 2.1. Production and Characterization of Spheroids Based on HER2-Overexpressing Cell Lines

HER2-overexpressing human ovarian adenocarcinoma cells, SKOV-3 (ATCC number HTB-77), and SKOV-kat cells (SKOV-3 transfected with the gene of red fluorescent protein TubroFP635 (Katushka, λex 588 nm, λem 635 nm)), were used [7].

Cells were cultured in McCoy’s 5A medium containing 1.5 mM glutamine (HyClone, Logan, UT, USA) and 10% (v/v) fetal calf serum (HyClone, Logan, UT, USA) at 37 °C in 5% CO_2_. For passaging, cells were detached with Versene solution (PanEco, Moscow, Russia).

Spheroids were produced in 96-well Ultra-Low-Attachment round-bottom plates (Corning, New York, NY, USA). SKOV-kat cells were seeded at concentrations of 200, 500, and 1000 cells/well and spheroids’ formation was monitored for 14 days in order to estimate their growth dynamics. Images of the spheroids were obtained by phase contrast microscopy using an Axiovert 200 inverted microscope with an EC PlanNeofluar 10×/0.3 objective lens (Carl Zeiss, Oberkochen, Germany). Volumes of individual spheroids (V, μm^3^) were calculated according to the equation of the ellipsoid volume:V = a × b^2^/2,(1)
where *a* is the larger diameter (μm) and *b* is the smaller diameter (μm).

Morphological study of SKOV-kat spheroids was performed by the following protocol. Growth medium was aspirated. After that, spheroids were dehydrated in ethanol of an increasing concentration (70%, 80%, 95%) for 5 min against three changes of each solution, then for 10 min in a 1:1 mixture of 95% ethanol/dimethylbenzene, and then in pure dimethylbenzene. Dehydrated spheroids were enclosed into a paraffin block. Then, 7–10 µm thick sections were obtained using a Microm HM 325 microtome (Thermo Fischer Scientific, Waltham, MA, USA). Sections were placed on highly adhesive Menzel-Glaser Superfrost ultra plus glass slides (Thermo Fischer Scientific, Waltham, MA, USA) and dried. Sections were deparaffinized by incubation for 2 min in pure dimethylbenzene and 96% ethanol against two changes of each solvent, and washed for 2 min in 70% ethanol, and then in distilled water. Fluorescent staining of spheroid sections was performed in the dark. Slides were incubated for 5 min in sodium acetate buffer (26.5 mL 0.2 M sodium acetate + 73.5 mL 0.2 M acetic acid, pH 4.2). Then, a freshly prepared acridine orange (PanEco, Moscow, Russia) solution in sodium acetate buffer (0.5 mg/mL) was applied onto the sections and incubated for 15 min and then washed 2 times with acetate buffer for 5 min and dried). Images of the spheroid sections were obtained using an Axio Observer Z1 LSM 710 NLO/Duo laser scanning microscope (Carl Zeiss, Oberkochen, Germany) with an EC Plan-Neofluar 20×/0.5 objective lens. For visualization of acridine orange and Katushka fluorescent protein, a 458 nm laser and 594 nm laser were used to excite fluorescence, respectively, and fluorescent signal was detected in the ranges of 505–552 nm and 600–740 nm, respectively.

For verification of the HER2 expression in spheroids, they were disaggregated with TrypLE solution (Thermo Fischer Scientific, Waltham, MA, USA) for 20 min at 37 °C in 5% CO_2_. The resulting cell suspension was incubated with HER2-specific antibodies labeled with fluorescein isothiocyanate (FITC)(Thermo Fischer Scientific, Waltham, MA, USA, Cat # AHO0918), according to the manufacturer’s recommendations, and then analyzed by flow cytometry using a FACSCalibur instrument (Becton Dickinson Franklin Lakes, NJ, USA). FITC fluorescence was excited with 488 nm argon laser, and the signal was detected in the range of 515–545 nm.

### 2.2. Production of the Targeted Toxin, DARPin-LoPE

DARPin-LoPE was produced in *Escherichia coli* BL21(DE3) cells transformed with the pDARP-LoPE plasmid containing the gene of DARPin-LoPE under T7 promoter control. Purification of DARPin-LoPE was performed by metal chelate affinity chromatography, as described in [8].

### 2.3. DARPin-LoPE and BSA Fluorescent Labeling

The DARPin-LoPE and bovine serum albumin (BSA) proteins were labeled with FITC or DyLight650 NHS Ester amine-reactive dyes (ThermoFisher Scientific, Waltham, MA, USA), which bind to the primary amino groups of the protein. For the reaction, proteins were exchanged into borate buffer (400 mM H_3_BO_3_, 70 mM Na_2_B_4_O7, pH 8.0) by gel filtration using a PD SpinTrap G-25 column (GE Healthcare, Chicago, IL, USA) according to the manufacturer’s recommendations. Then, the proteins were incubated with a 30-fold molar excess of FITC or with a 7-fold molar excess of DyLight650 at room temperature in the dark for 1 h. Unbound dye was removed by gel filtration using a PD SpinTrap G-25 column equilibrated with phosphate-buffered saline (PBS, pH 7.4) (Sigma-Aldrich, St. Louis, MO, USA).

### 2.4. Evaluation of DARPin-LoPE, Doxorubicin, and BSA Penetration into a Spheroid

Cells were seeded in 96-well Ultra-Low-Attachment round-bottom plates (Corning, New York, NY, USA), 2000 cells/well, and incubated for 4 days to form spheroids. Spheroids were then incubated for 2, 12, or 24 h in growth medium containing 1 μM FITC- or DyLight650-labeled DARPin-LoPE or 1 μM FITC-labeled BSA or 10 μM doxorubicin. Then, spheroids were washed twice with PBS and fixed with 4% formaldehyde solution in PBS for 15 min in the dark. Images of spheroids were obtained using an Axio Observer Z1 LSM 710 NLO/Duo laser scanning microscope (Carl Zeiss, Oberkochen, Germany) with an EC Plan-Neofluar 20×/0.50 objective lens. FITC and doxorubicin fluorescence was excited using a 488 nm argon laser, and signal detection was performed in ranges of 500–560 nm and 517–614 nm, respectively. Fluorescence of Katushka fluorescent protein and DyLight650 dye was excited using 594 nm or 633 nm helium-neon lasers, respectively, and signal was detected in the ranges of 600–740 nm and 653–730 nm, respectively. To estimate the agent’s penetration into spheroids by flow cytometry, spheroids were disaggregated to a single-cell suspension according to the abovementioned protocol, and subsequently analyzed using a FACSCalibur instrument (Becton Dickinson, Franklin Lakes, NJ, USA). Doxorubicin and DyLight650 fluorescence were excited using a 488 nm argon laser or 635 nm helium-neon laser, respectively, and the signal was detected in the ranges of 564–606 nm and 653–669 nm, respectively.

### 2.5. Cytotoxicity Analysis of DARPin-LoPE and Doxorubicin

Cytotoxicity of DARPin-LoPE and doxorubicin against spheroids was compared to the cytotoxicity against the monolayer culture. 

In order to evaluate cytotoxicity against the monolayer culture, 2000 cells/well were seeded into 96-well tissue culture plate (Corning, New York, NY, USA) and cultured overnight. Then, the growth medium in the wells was exchanged with fresh medium containing 10^−5^ to 10^3^ nM DARPin-LoPE or 0.027 to 20 μM doxorubicin. Cell viability was estimated using the methylthiazoletetrazolium (MTT) assay [9] after 72 h incubation. The medium in the wells was exchanged with the fresh medium, containing 0.5 mg/mL MTT (Alfa Aesar, United Kingdom), and cells were incubated for 4 h. After that, formazan crystals were dissolved in dimethyl sulfoxide (PanEco, Moscow, Russia). The optical density was measured at 570 nm using a Synergy MX microplate reader (BioTek, Winooski, VT, USA). The relative cell viability was calculated as the percentage of mean optical density in the wells with treated cells to the mean optical density in the wells with untreated cells.

Cytotoxicity of DARPin-LoPE and doxorubicin against the spheroids was estimated by the following protocol. Cells were seeded in 96-well Ultra-Low-Attachment round-bottom plates (Corning, New York, NY, USA), 1000 cells/well, and incubated for 4 days to form spheroids. Then, the growth medium in the wells was replaced with fresh medium, containing DARPin-LoPE or doxorubicin in various concentrations, and spheroids were incubated for 72 h. Images of spheroids were obtained every day using an Axiovert 200 inverted microscope with an EC PlanNeofluar 10×/0.3 objactive lens (Carl Zeiss, Oberkochen, Germany). Cytotoxicity of the therapeutic agents against spheroids was evaluated according to the spheroid volume on the final day of incubation in the presence of the agents: Relative viability was calculated as a percentage of the mean volumes of treated to untreated spheroids.

The value of the half maximal inhibitory concentration (IC_50_)was calculated using GraphPad Prism software (GraphPad Software, version 6.0 for Windows, San Diego, CA, USA, 2012) using the nonlinear regression using a four-parameter dose–response model.

## 3. Results

### 3.1. Production and Characteristics of SKOV-kat Spheroids

To obtain spheroids of human ovarian adenocarcinoma, we used 96-well round-bottom ultra-low-attachment plates, which were previously successfully tested for the production of human breast adenocarcinoma spheroids [10]. Spherical conglomerates of cells with a clearly defined border (spheroids) were formed on the fourth day after seeding the human ovarian adenocarcinoma SKOV-3 or SKOV-kat cells in the plate (Figure 1A).

To estimate the growth dynamics of spheroids, we monitored their development at different seeding densities in the plate wells. Figure 1B shows the averaged growth curves of SKOV-kat spheroids. At a low-density seeding (200 cells/well), the growth dynamics are close to linear, while at higher densities (500 and 1000 cells/well), the growth curve has a more complex nature approximated by a sigmoid. Such dynamics are considered inherent in the growth of tumors in vivo [11]. Spheroid growth slowdown over time can be caused by the increasing role of hypoxia and insufficient nutrient delivery into the spheroid as its size increases, which ultimately leads to the formation of a necrotic core in the spheroid [12]. Since the effect of therapeutic agents should be estimated in a phase of active growth of spheroid, subsequent experiments were carried out before the eighth day of spheroid growth. To study the morphology of the SKOV-kat spheroids at this time, the spheroids’ sections were stained with acridine orange, which binds to double-stranded nucleic acids (Figure 1C). Homogeneous fluorescence of acridine orange (green) and Katushka protein (red) throughout the depth of the spheroid indicates the viability of cells throughout the spheroid and the absence of a necrotic core in its center.

A key feature of the SKOV-kat cell line is the overexpression of the HER2 receptor. As shown by flow cytometry of cell suspensions after staining with FITC-labeled HER2-specific antibody, SKOV-kat cells preserve the HER2 receptor in both 2D monolayer and 3D spheroid culture (Figure 1D).

### 3.2. Depth and Dynamics of Penetration of DARPin-LoPE, BSA, and Doxorubicin into the Spheroid

The well-documented relative resistance of a tumor in vivo to the action of anticancer drugs, compared with a monolayer cell culture [13], is caused by a combination of factors, one of which is insufficient accumulation of the drug in the tumor as a result of its poor penetration into the tumor mass. This, in turn, can be determined both by the features of the three-dimensional structure of the tumor tissue and by the properties of the agent itself (diffusion ability, affinity to cell-surface targets, rate of internalization, and mechanism of action). To evaluate the influence of the size (molecular weight) of the antitumor agent on its penetration into the depth of the tumor, we incubated SKOV-kat spheroids in the presence of antitumor agents with different molecular weights. As a high-molecular-weight agent, we used the FITC-labeled recombinant targeted toxin, DARPin-LoPE (Mr 42 kDa), composed of the HER2-specific DARPin and low-immunogenic fragment of *Pseudomonas* exotoxin A (denoted by “LoPE”) [8]. DARPins (Designed Ankyrin Repeat Proteins) present a class of non-immunoglobulin scaffold proteins, which are actively used for tumor targeting due to their small size, high stability, and high-yield production in bacteria [14,15]. Like its previous analogues [7,16,17] DARPin-LoPE showed a high specific cytotoxicity against HER2-positive cells in vitro [8]. This variant of the targeted toxin is characterized by reduced immunogenicity due to the mutation of immunodominant epitopes of human B lymphocytes [18], which represents prospects for its clinical use. Doxorubicin, which is widely used in clinical practice and has its own orange-red fluorescence (λex 488 nm, λem 517–614 nm), was chosen as a low-molecular-weight antitumor agent. As shown in Figure 2, DARPin-LoPE is characterized by slow accumulation in the cells of the surface layers of the spheroid with penetration to a depth of about 50 to 70 μm only (at a spheroid diameter of about 300 nm) during 24 h of incubation (Figure 2D). The low-molecular-weight agent, doxorubicin, also shows gradient accumulation throughout the depth of the spheroid, but its penetration is much more intensive as compared to DARPin-LoPE under similar conditions (Figure 3).

For targeted agents characterized by a high affinity for their molecular target, a so-called “binding site barrier” effect may take place, which is attributed to the stable binding of the agent to the tumor cell antigens along the periphery of the tumor. This hinders further penetration of such agents deep into the tumor tissue and, as a result, reduces their effectiveness [19]. To evaluate the possible effect of this phenomenon on the penetration of the HER2-specific targeted toxin, DARPin-LoPE, we also visualized the penetration of FITC-labeled bovine serum albumin, a protein with a molecular weight of the same order (66 kDa), but with no specificity to HER2. Figure 4 shows that BSA penetrates into the SKOV-kat spheroid poorly, similar to DARPin-LoPE. Thus, the ineffective penetration of DARPin-LoPE seems to be mainly due to the high molecular weight of this agent. It is interesting to note that K. Winner and colleagues [20] compared the penetration of the HER2-specific antibody, pertuzumab, and a non-specific antibody of the same isotype and molecular weight into SKOV3.ip1 spheroids: The HER2-specific pertuzumab did not penetrate well even with prolonged (24 h) incubation, while non-specific antibody quickly and efficiently diffused into the spheroid. Thus, spheroids of different ovarian carcinoma cell lines exhibit various degrees of diffusion of non-specific molecules, which can be determined by peculiar features of a three-dimensional spheroid structure, such as density and composition of cell-to-cell contacts. The diffusion of target-specific molecules is also influenced by the dynamics of the interaction with their targets on the cell surface.

Due to the optical characteristics of biological tissue (in particular, strong absorption and scattering of blue and green light), the penetrating power of visible light varies quite a lot when moving from the blue to the red region of the spectrum [21]. Using the SKOV-kat spheroids expressing fluorescent protein, Katushka (635 nm), we demonstrated the possibility of the imaging of red fluorescence from the depth of spheroids with a diameter of 300 to 400 μm by an objective lens with a numerical aperture of 0.5 (see Figure 2, Figure 3 and Figure 4). In this regard, in order to avoid the misinterpretation of results on the penetration of DARPin-LoPE and BSA protein molecules labeled with green dye FITC, we further evaluated the penetration of DARPin-LoPE labeled with the red fluorescent dye, DyLight650, into spheroids obtained from cells of the non-fluorescent (parental) SKOV-3 line (Figure 5). In this case, an accumulation dynamics similar to that of DARPin-LoPE-FITC was observed: The depth of penetration did not exceed 50 to 70 μm after 24 h of incubation (for details, see z-stack in Appendix A).

The obtained data are also confirmed by flow cytometry. The SKOV-3 spheroids incubated for various times in the presence of DyLight650-labeled DARPin-LoPE or doxorubicin were disaggregated and analyzed with a flow cytometer. A slight shift in cell populations along the fluorescence intensity in the DyLight650 channel over time indicates a low accumulation of labeled DARPin-LoPE in the cells of spheroids (Figure 6A), while the accumulation of doxorubicin is much more significant (Figure 6B).

### 3.3. Cytotoxicity of DARPin-LoPE and Doxorubicin

To study the cytotoxicity of the agents against the monolayer cell culture, the standard MTT assay was performed [9]. To estimate the effect of the agents on the growth of spheroids, the dynamics of the spheroids’ volume upon incubation with the studied agent were analyzed. In preliminary experiments, we showed that the assessment of the relative viability of cells based on the MTT data and size of the spheroids gives well-correlated results (Appendix A).

The results show that SKOV-kat cell viability in the monolayer was significantly reduced already at picomolar concentrations of the targeted toxin, DARPin-LoPE (IC_50_ 0.1 nM), while significant inhibition of spheroid growth was not achieved even at DARPin-LoPE concentrations of about 100 nM (IC_50_ 2600 nM, Figure 7A). The severity of the effect of doxorubicin also differed between the monolayer culture and spheroids, which corresponds to its gradient penetration into spheroids (Figure 3), but the differences found were not as significant as in the case of the targeted toxin (Figure 7B): the calculated IC_50_ values of doxorubicin for cells in the monolayer and in spheroids differed by less than an order (160 nM and 730 nM for monolayer and spheroids, respectively).

It is worth noting that the penetration efficacy should also be considered when analyzing mechanisms of cell death. For instance, staining of living spheroids with traditionally used Annexin V to distinguish cells dying via apoptosis can be incorrectly interpreted since Annexin V (~36 kDa) penetration is most likely limited (see Appendix A for details). 

## 4. Discussion

The effectiveness of therapeutic agents’ delivery to the tumor is among the extremely important criterions determining the success of their clinical use. Thus, its requires thorough research in each particular case. The size of the molecule or supramolecular complex largely determines its pharmacokinetics and biodistribution, as well as the effectiveness of extravasation and further diffusion in the tissue. In general, larger complexes are characterized by a longer blood circulation, but weak penetration from the vessel into the interstitium. Smaller agents, on the contrary, are able to extravasate and diffuse into the tissue more effectively, but are quickly removed from the bloodstream by the liver, kidneys, and spleen, and their concentration in the blood is often insufficient for effective accumulation in the tumor. These patterns are shown for agents of different nature: In particular, for polymer micelles [22,23], gold [24], and oxide silicon [25] nanoparticles, antibodies, and their fragments [26,27], when they are systemically administered to a tumor-bearing animals, as well as in vitro using avascular tumor models (tumor spheroids or extracted tumor nodes). It is worth noting that the absolute values of the penetration rates of various agents differ significantly. For example, as it is reported, 30 nm polymeric nanoparticles homogenously penetrated to a depth of 100 μm from the vessel into a xenograft model of human breast cancer, while the penetration depth of 100 nm particles did not exceed 40 μm [23]. In another study, 25 nm-sized polymer micelles were shown to penetrate in a xenograft breast cancer model to a depth of 40 μm from the vessel, while 60 nm particles diffused only to 20 μm. Moreover, surface functionalization of these micelles with the targeting molecule led to an additional decrease in penetration depth due to the “binding site barrier” effect [22]. Such differences may be caused by a variety of factors affecting the penetration of the agent into the tumor tissue after systemic administration, including both features of the tumor biology (cell–cell junctions, density, vascularization, stiffness of extracellular matrix, etc.) and physicochemical properties of the agent (size, surface charge, functionalization, etc.). For low-molecular-weight antitumor agents, such as doxorubicin, the penetration depth is largely determined by their chemical properties and also varies greatly, but usually does not exceed 100 μm from the vessel [28,29].

In this study, we used human ovarian carcinoma spheroids to study the effectiveness of the targeted protein toxin. Spheroids are quite widely used as a model of an avascular tumor, since they allow the influence of the whole-body mechanisms of transport and biodistribution of the therapeutic agent to be eliminated and the penetration efficiency of the agent into a specific type of tumor at the initial stage of the potential drug testing to be estimated [5]. In addition, in the case of ovarian carcinoma, microscopic avascular spheroids present one of the stages of the metastatic process in the abdominal cavity [30], and the best way to deliver agents to such tumors is considered to be intraperitoneal injection and drug distribution in the peritoneal fluid [20]. Therefore, the spheroid model in vitro gains obvious clinical relevance.

We evaluated the cytotoxic action of two antitumor agents, the high-molecular-weight HER2-specific targeted toxin, DARPin-LoPE, and the low-molecular-weight chemotherapy drug, doxorubicin, against the tumor spheroids, SKOV-kat, and compared it with the effect on monolayer culture. The analysis showed a higher resistance of spheroids to the high-molecular-weight agent than to the low-molecular-weight agent. It should be noted that the transition from 2D to 3D culture conditions may have different effects on the sensitivity of tumor cells to the action of antitumor agents. In relation to targeted agents specific to the EGFR (HER) family proteins, there are reported cases of both a decrease and an increase in efficacy in 3D models, as compared to 2D. Thus, in [31], an increase in the resistance of HER2-overexpressing lung carcinoma spheroid cells to the action of a photoimmunotherapy agent was revealed as compared with the monolayer culture. Additionally, in [32], on the contrary, a more pronounced sensitivity of the HER2-overexpressing ovarian and breast carcinoma spheroids to the action of the HER2-specific monoclonal antibody, trastuzumab, was shown. These differences can be associated both with the efficiency of penetration of an agent into the spheroid, and with the mechanism of the agent’s action in combination with the change in the biology of the target receptors during the transition from 2D to 3D. In our case, the significant resistance of the SKOV-kat spheroids to the action of a high-molecular-weight targeted toxin is associated, at least partially, with its insufficient penetration into the depth of the spheroid, acting only on the cells of the surface layers. The problem of poor penetration into ovarian carcinoma has been indicated for a number of agents of different natures: The HER2-specific antibody, pertuzumab [20], targeted micelles [33], and silica nanoparticles [34].

We created a number of recombinant toxins of various mechanisms of action for targeted therapy of HER2-positive tumors: Targeted toxins based on *Pseudomonas* exotoxin A [7,16], and immunophotosensitizers based on KillerRed [35] and miniSOG [36,37] phototoxic proteins. These agents showed extremely high and selective toxicity against HER2-overexpressing tumor cells both on tumor cell cultures in vitro and on xenograft tumor models in vivo. However, the concentrations of agents necessary to achieve a therapeutic activity in vivo are significantly higher than the effective in vitro concentrations [7,16]. The higher the concentration of the toxin administered in vivo, the more evident side toxicity expected, as well as the immunogenicity.

In this regard, approaches are currently being actively developed to improve the delivery and penetration of therapeutic agents into tumors. For instance, a promising strategy for optimizing the biodistribution and tumor accumulation of agents during systemic (intravenous) administration is the development of size shrinkable drug delivery nanosystems. The idea is that the larger sized complex firstly achieves blood concentrations sufficient for its extravasation in the tumor, where the stimuli-sensitive release of smaller active agents occurs, which penetrate deeper into the tumor [38]. A similar approach has shown prospects for ovarian cancer treatment [39]. Another concept tested on ovarian carcinoma is to use cell-penetrating/tumor-targeting peptides for a better agent penetration into tumor/tumor cells [40,41], as well as cell-mediated delivery [34]. A further group of approaches involves the impact on the tumor itself, in particular, the destruction of the extracellular matrix or the decrease in its production by stromal fibroblasts [42,43], and the temporary destruction of cell–cell junctions [44]. The latter was successfully tested using a recombinant agent based on capsid proteins of human adenovirus serotype 3 (Ad3), specific to desmoglein 2 (JO). The reversible opening of tight junctions under the JO action led to an increase in the antitumor effect of the drugs co-administered with JO [45,46]. One more concept in this area is based on normalization of an aberrant tumor vasculature. Well-vascularized tumors obviously possess higher grades of drug delivery efficiency. However, in most cases, tumor vasculature is disordered and leaky, causing elevated interstitial fluid pressure and preventing drug extravasation and penetration to tumor depth [47]. Temporary normalization of the vasculature with anti-angiogenic therapy aims to restore the normal tissue fluid convection, thus being a promising approach to improve drug penetration efficiency [48]. There are several ongoing clinical trials in this area, however, the results are contradictory [49]. It is worth noting that shrinkage of the pore size in the vessel walls under anti-angiogenic treatment is possible and can prevent extravazation of high-molecular weight agents [50], so the dosing must be carefully conducted.

Thus, optimization of the delivery and accumulation of a therapeutic agent in a tumor, as well as solving the problem of drug penetration throughout the entire thickness of the tumor, is extremely important for the successful treatment of solid tumors and may have a particular practical importance for the clinical promotion of high-molecular targeted agents [51].

## Figures and Tables

**Figure 1 pharmaceutics-11-00219-f001:**
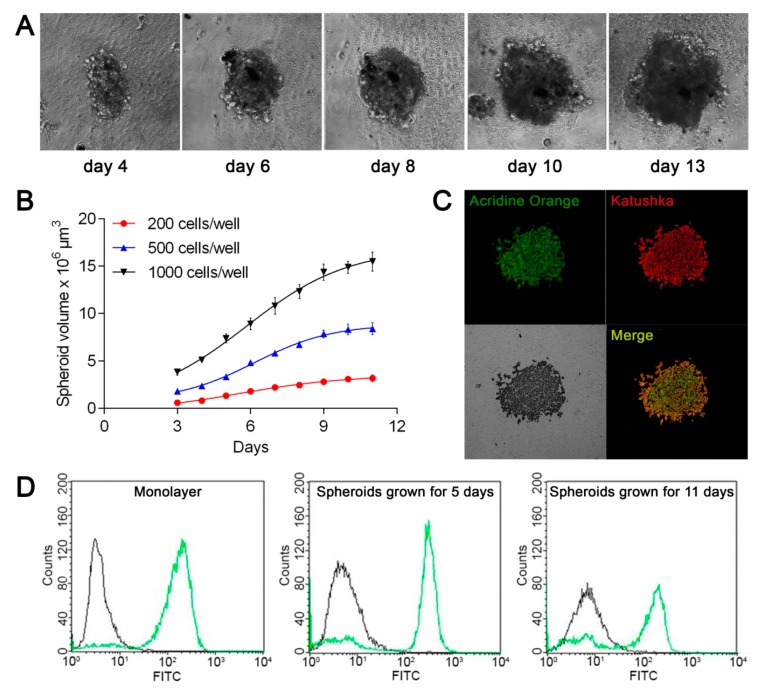
Production and characterization of human ovarian adenocarcinoma spheroids SKOV-kat. (**A**) SKOV-kat spheroid morphology on different days of growth after seeding 500 cells/well. Images size 550 × 550 μm; (**B**) Growth curves of SKOV-kat spheroids obtained at different seeding densities. The day when the cells were seeded in plate was set as day “0”; (**C**) confocal images of a SKOV-kat spheroid section obtained on the eighth day of growth after seeding 2000 cells/well, and stained with acridine orange. Fluorescence of acridine orange and Katushka fluorescent protein is presented in green (λex 458 nm, λem 505–552 nm) and red (λex 594 nm, λem 600–740 nm), respectively. Image size is 707 × 707 μm; (**D**) analysis of the human epidermal growth factor receptor 2 HER2 expression by SKOV-kat cells in monolayer culture (on the left) and in spheroids on the 5th (in the center) and on the 11th (on the right) days of growth. The distributions of SKOV-kat cells according to fluorescence intensity in the fluorescein isothiocyanate (FITC) channel after staining with FITC-labeled HER2-specific antibodies (green curve) or antibodies of isotypic control (black curve) are presented.

**Figure 2 pharmaceutics-11-00219-f002:**
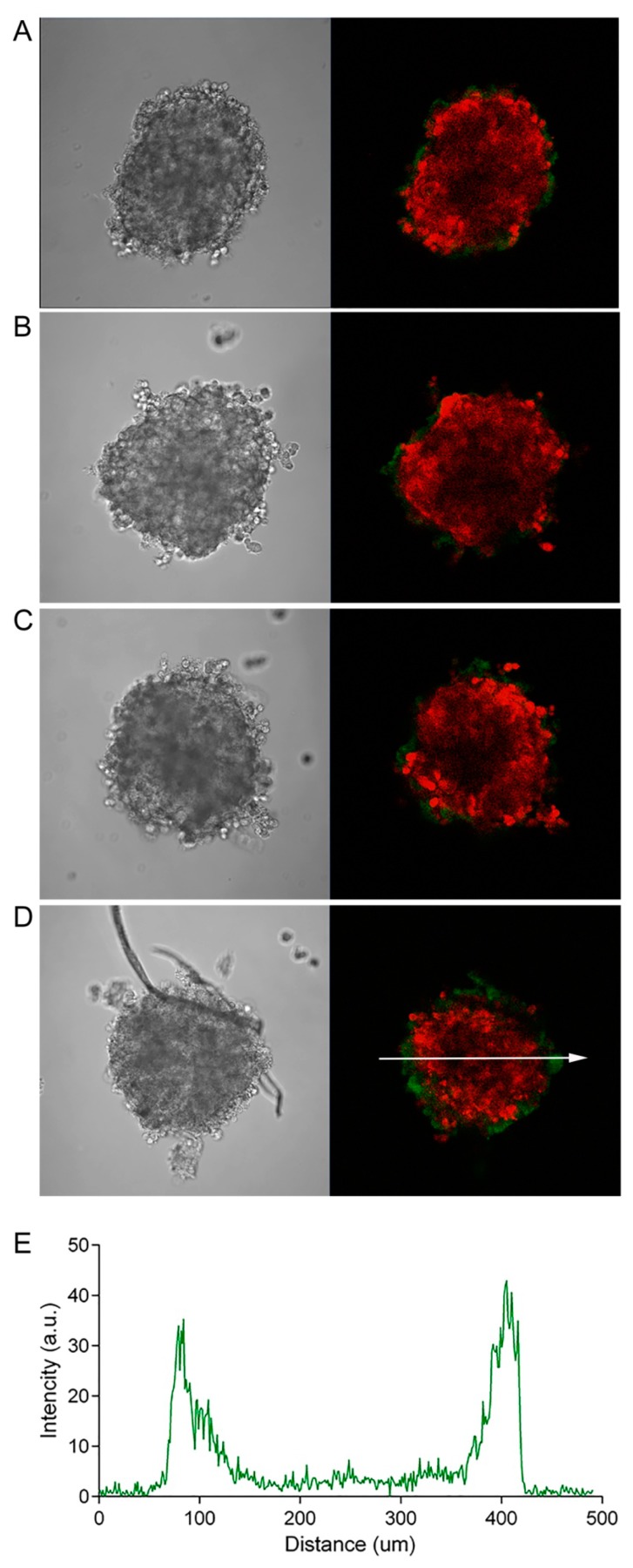
Dynamics of the penetration of the FITC-labeled recombinant targeted toxin, DARPin-LoPE, into SKOV-kat spheroids. The SKOV-kat spheroids (red) were analyzed by confocal microscopy in control (no DARPin-LoPE, **A**) and after incubation with FITC-labeled DARPin-LoPE (green) for 2, 12, or 24 h (**B**–**D**, respectively). Image size is 708 × 708 μm. The fluorescence intensity profile (**E**) shows the fluorescence intensity of the FITC-labeled, DARPin-LoPE, along the randomly positioned arrow in (**D**).

**Figure 3 pharmaceutics-11-00219-f003:**
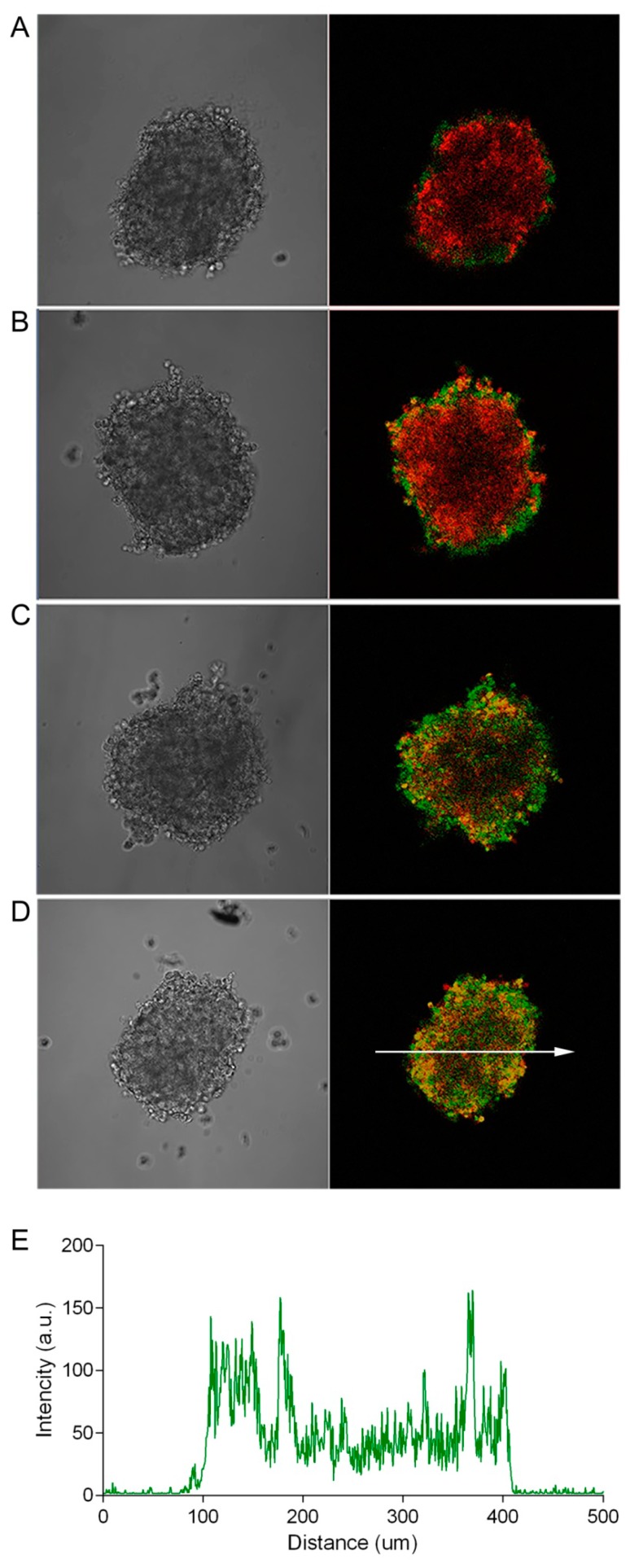
Dynamics of the penetration of doxorubicin into SKOV-kat spheroids. The SKOV-kat spheroids (red) were analyzed by confocal microscopy in control (no doxorubicin, **A**) and after incubation with doxorubicin (green) for 2, 12, or 24 h (**B**–**D**, respectively). Image size is 708 × 708 μm. The fluorescence intensity profile (**E**) shows the fluorescence intensity of doxorubicin along the randomly positioned arrow in (**D**).

**Figure 4 pharmaceutics-11-00219-f004:**
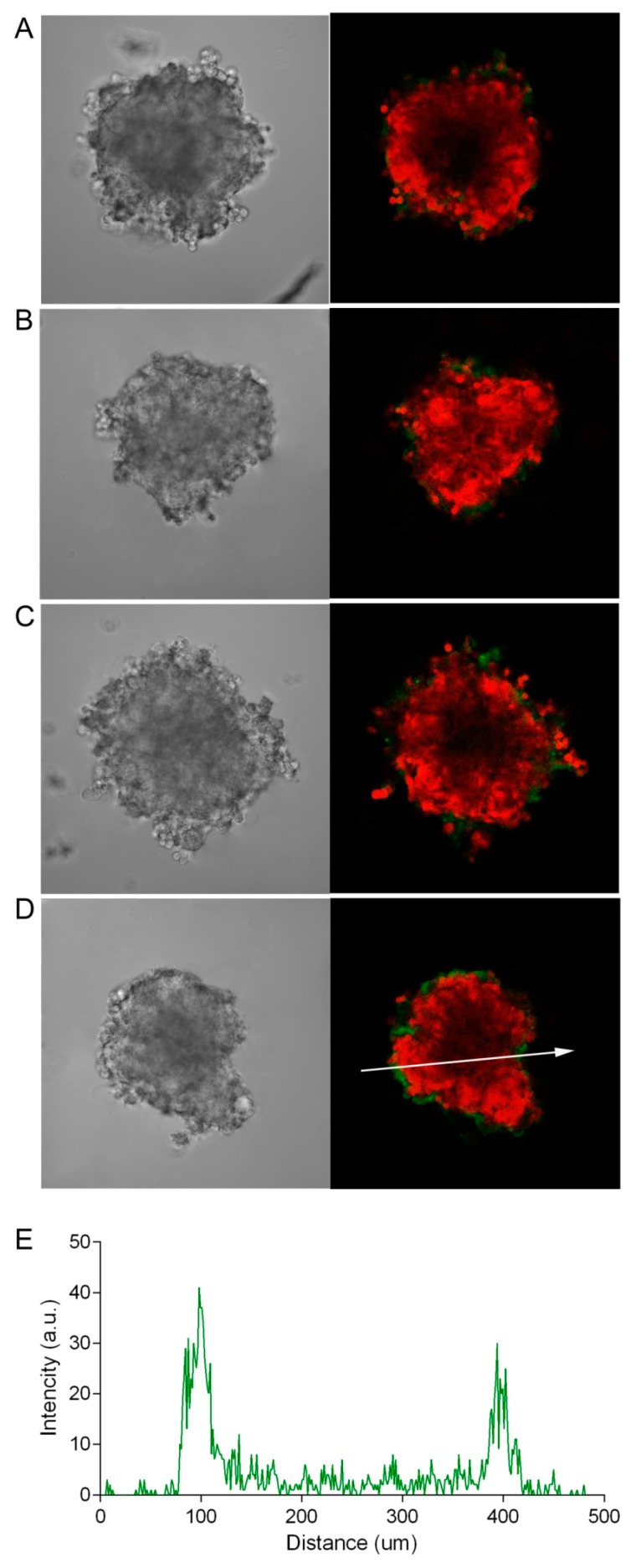
Dynamics of the penetration of FITC-labeled bovine serum albumin (BSA) into SKOV-kat spheroids. The SKOV-kat spheroids (red) were analyzed by confocal microscopy in control (no BSA, **A**) and after incubation with FITC-labeled BSA (green) for 2, 12, or 24 h (**B**–**D**, respectively). Image size is 698 × 698 μm. The fluorescence intensity profile (**E**) shows the fluorescence intensity of FITC-labeled BSA along the randomly positioned arrow in (**D**).

**Figure 5 pharmaceutics-11-00219-f005:**
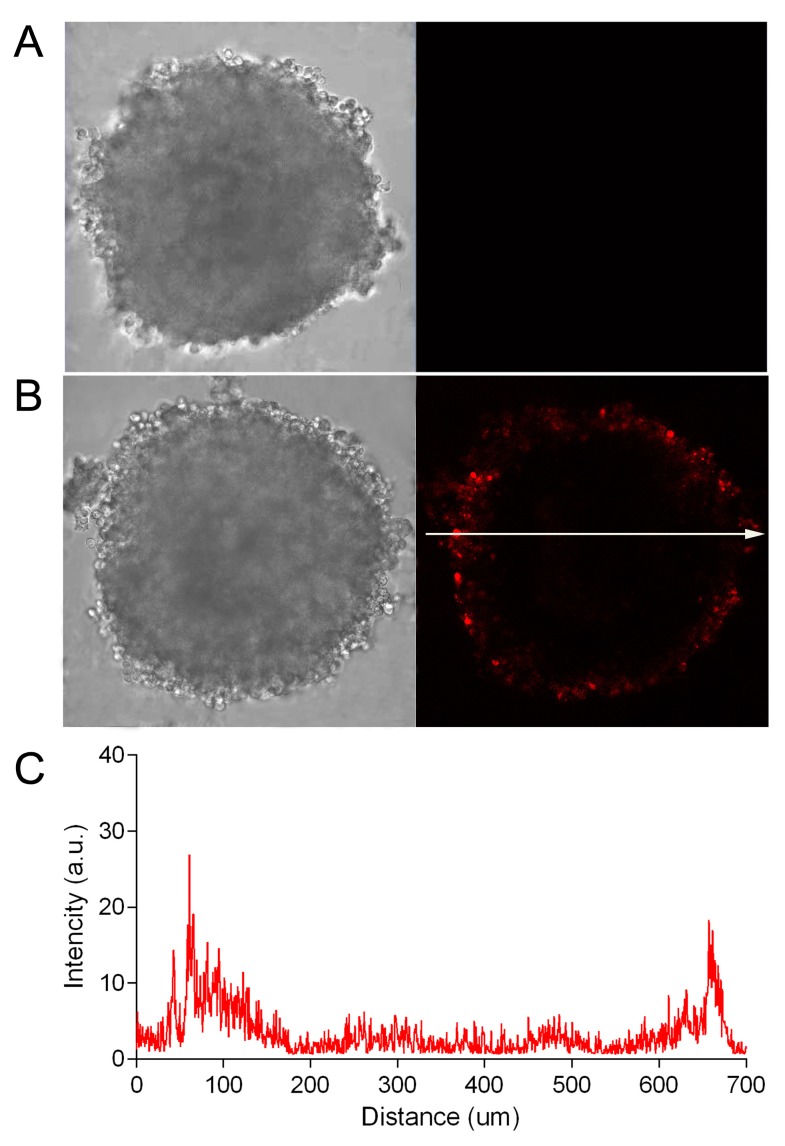
Dynamics of the penetration of DyLight650-labeled recombinant targeted toxin DARPin-LoPE into SKOV-3 spheroids. The SKOV-3 spheroids were analyzed by confocal microscopy in control (no DARPin-LoPE, **A**) and after incubation with DyLight650-labeled DARPin-LoPE (red) for 24 h (**B**). Image size is 708 × 708 μm. The fluorescence intensity profile (**C**) shows the fluorescence intensity of DyLight650-labeled DARPin-LoPE along the randomly positioned arrow in (**B**).

**Figure 6 pharmaceutics-11-00219-f006:**
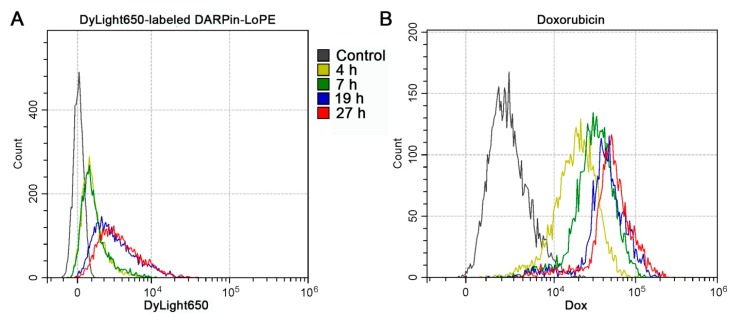
Dynamics of the accumulation of DARPin-LoPE and doxorubicin in SKOV-3 spheroids. The distribution of SKOV-3 cells according to the fluorescence intensity after incubation of SKOV-3 spheroids with DARPin-LoPE-DyLight650 (**A**) or doxorubicin (**B**) for various times with their subsequent disaggregation and analysis of cell suspensions by flow cytometry.

**Figure 7 pharmaceutics-11-00219-f007:**
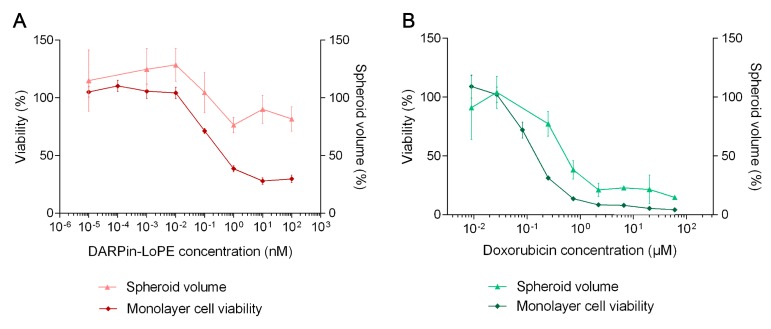
Cytotoxicity of recombinant targeted toxin DARPin-LoPE (**A**) and doxorubicin (**B**) against SKOV-kat culture: the relative viability of cells in the monolayer according to the MTT assay (left Y axis) and the relative volume of spheroids (right Y axis) after incubation with the targeted toxin or doxorubicin for 72 h. Data are presented as mean ± SEM (*n* = 6).

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
