# Peer review of "Penetration Efficiency of Antitumor Agents in Ovarian Cancer Spheroids: The Case of Recombinant Targeted Toxin DARPin-LoPE and the Chemotherapy Drug, Doxorubicin"

_pharmaceutics, 2019, doi:10.3390/pharmaceutics11050219_

Round 1

Reviewer 1 Report

In their manuscript ”Size-dependent penetration efficiency of antitumor agents in ovarian cancer spheroids”, Sokolova et al. compare penetration and efficacy of doxorubicin with the HER2-targeting agent DARPin-LoPE. While this is work is of interest to the readership of Pharmaceutics, some points should be addressed.

1)

The authors should differentiate between targeted drugs of low-molecular weight (e.g. kinase inhibitors) and antibody-based targeted drugs. In the case of receptor tyrosine kinases, both strategies are already used clinically.

2)

The authors should discuss under which circumstances diffusion is limiting for cancer drug – and when this obstacle is not as pronounced (highly vascularised tumours). They might also discuss how antiangiogenic drugs might interfere with high-molecular weight drug efficacy.

3)

Figure 7: I suggest to plot dose-response curves rather than indivudal bar graphs and also calculate IC50 values to allow numeric comparison.

4)

It is apparent that even for doxorubicin, there is a concentration gradient in the spheroids, also reflected by reduced toxicity (which is why IC50 values are of particular relevance). This has to be discussed. Could the authors supply a model of molecule size and achievable concentrations in spheroid cores?

5)

The authors need to demonstrate (e.g. by staining for apoptosis markers) that there are differences in cytotoxicity (periphery of spheroids vs core of spheroids) for doxorubicin as compared to DARPin-LoPE. Otherwise, there is no mechanistic link proven between lowered toxicity of DARPin-LoPE in spheroids and its penetration.

Author Response

Dear Reviewer,

We would like to express our sincere appreciation for your careful attention to our manuscript and for the suggested improvements and valuable comments. We have revised the manuscript according to your remarks. Please find below the detailed description of the revisions (Reviewer’s comments are presented in bold, our answers follow in plain text).

Comments and Suggestions for Authors:

English language and style are fine/minor spell check required 

The English was carefully revised.

In their manuscript ”Size-dependent penetration efficiency of antitumor agents in ovarian cancer spheroids”, Sokolova et al. compare penetration and efficacy of doxorubicin with the HER2-targeting agent DARPin-LoPE. While this is work is of interest to the readership of Pharmaceutics, some points should be addressed.

1)      The authors should differentiate between targeted drugs of low-molecular weight (e.g. kinase inhibitors) and antibody-based targeted drugs. In the case of receptor tyrosine kinases, both strategies are already used clinically.

We corrected the Introduction section to make this point more clear (lines 57-60).

2)      The authors should discuss under which circumstances diffusion is limiting for cancer drug – and when this obstacle is not as pronounced (highly vascularised tumours). They might also discuss how antiangiogenic drugs might interfere with high-molecular weight drug efficacy.

We thank the Reviewer for the valuable suggestion to enrich the discussion, the appropriate text was added (lines 413-422).

3)      Figure 7: I suggest to plot dose-response curves rather than individual bar graphs and also calculate IC50 values to allow numeric comparison.

Figure 7 and its caption were modified. The IC50 values (the concentration of the therapeutic agent that causes a decrease in monolayer cell viability or in spheroid volume by 50% relative to the control) are given in paragraph 3.3 (lines 313-321).

4)      It is apparent that even for doxorubicin, there is a concentration gradient in the spheroids, also reflected by reduced toxicity (which is why IC50 values are of particular relevance). This has to be discussed. Could the authors supply a model of molecule size and achievable concentrations in spheroid cores?

We thank the Reviewer for the valuable note regarding the doxorubicin penetration gradient, the corresponding corrections were made (lines 237-240, 317-318).

The creation of the mathematical model was beyond the scope of our study.

To the present moment, a simulation of the tumor uptake is presented in a large number of published works. Some recent works are listed below:

-          Steuperaert et al. A 3D CFD model of the interstitial fluid pressure and drug distribution in heterogeneous tumor nodules during intraperitoneal chemotherapy. Drug Deliv (2019) 26(1):404-415. doi: 10.1080/10717544.2019.1588423.

-          Hubbard et al. Drug delivery in a tumour cord model: a computational simulation. R Soc Open Sci. (2017) 4(5):170014. doi:10.1098/rsos.170014.

-          Orcutt et al.  Molecular Simulation of Receptor Occupancy and Tumor Penetration of an Antibody and Smaller Scaffolds: Application to Molecular Imaging. Mol Imaging Biol (2017) 19: 656. doi:10.1007/s11307-016-1041-y.

-          Namazi et al. Mathematical Based Calculation of Drug Penetration Depth in Solid Tumors. Biomed Res Int (2016) 8437247. doi: 10.1155/2016/8437247.

-          Vasalou et al. A mechanistic tumor penetration model to guide antibody drug conjugate design. PLoS One. (2015). 10(3):e0118977. doi: 10.1371/journal.pone.0118977.

In addition to the size of the molecule and its diffusion coefficient, the blood concentration, tumor clearance due to internalization and vanishing, affinity to cell receptors, and side barrier effect are considered as the main factors influencing drug penetration to tumor. In worth noting, that the simulation results are sometimes contradictory when formulating the optimal combination of the parameters. So, not one of the model can be stated as validated and absolutely accurate in prediction.

5)      The authors need to demonstrate (e.g. by staining for apoptosis markers) that there are differences in cytotoxicity (periphery of spheroids vs core of spheroids) for doxorubicin as compared to DARPin-LoPE. Otherwise, there is no mechanistic link proven between lowered toxicity of DARPin-LoPE in spheroids and its penetration.

We are grateful to the Reviewer for this valuable comment. We have carried out such experiments in order to evaluate the cytotoxic effect of the agent in the spheroid spatially (on the periphery and in depth). For example, Figure R1 (in attached file) shows SKOV-3 spheroids incubated for 3, 4 or 5 days in the presence of the targeted toxin DARPin-LoPE, then stained with a mixture of Annexin V-FITC and Propidium iodide to differentiate the type of cell death, and visualized by confocal microscopy. The PE-based targeted toxins that we have been developing are characterized by induction of apoptosis in target cells, as we showed earlier for DARPin-LoPE (Proshkina et al., Molecular biology, 2017, doi:10.1134/s0026893317060140) and for similar targeted toxins (Sokolova et al., Journal of controlled release, 2016, doi: 10.1016/j.jconrel.2016.05.020; Sokolova et al., Oncotarget, 2017, doi: 10.18632/oncotarget.15833). A cell dying via apoptosis is typically stained with Annexin V-FITC (at the initial stages) or with both Annexin V-FITC and Propidium iodide (at the later stages of apoptotic death). We observed staining with Propidium iodide throughout the entire volume of the treated spheroids, while Annexin V-FITC was localized only on the spheroids surface. Moreover, this staining pattern did not depend on the time of incubation of the spheroids with the toxin. Apparently, it is impossible to interpret the results of such experiments correctly when working with living intact spheroids, because again the question about the penetration of dyes into the spheroid arises. Propidium iodide being a low- molecular-weight compound (~0.65 kDa), seems to penetrate well into the spheroid, while penetration of the Annexin protein (~36 kDa) is most likely difficult. In this regard, these results were not included in the article.

We absolutely agree with the Reviewer that the data we have obtained are not enough to establish a direct correlation between the decrease in the DARPin-LoPE toxicity against spheroids and its poor penetration. We corrected this statement appropriately (lines 28-29, 79-81, 386).

Reviewer 2 Report

The manuscript entitled “Size-dependent penetration efficiency of antitumor agents in ovarian cancer spheroids” reports how the HER2-specific targeted toxin DARPin-LoPE poorly penetrates into three-dimensional spheroids.

General comment: To me, the unique cell line used in this article shows a very epithetial phenotype likely making spheroids highly dense. It would have been interesting to look at whether more invasive cells like the SKOV-3 derivative, SKOV3.ip1, make the same kind of spheroids and whether DARPin-LoPE and/or BSA might show a better penetration in such spheroids. This is important since others (Winner et al.) have shown that high molecular weight antibodies might enter into this 3D cell model. More generally, it would have been interesting to compare different cell lines with different invasive phenotypes since chemotherapeutics are mostly used to treat advanced diseases.

Other comments are detailed below:

Comment 1:  In the introduction section, the authors state that “However, when testing such agents in vivo, a significant decrease in their effectiveness is revealed. If such agents further undergo clinical trials no superiority is often observed compared to conventional treatment”. Such assumptions should be carefully used since anti-Her2 therapeutic strategies have shown activities in clinic and demonstrated superiority compared to conventional intervention.

Comment 2: On Figure 6, the authors disaggregated DARPin-LoPE treated 3D spheroids and analyzed cells by flow cytometry. While in microscopy, the authors show a clear difference between the labeling of the cells located at the surface of the spheroids compared to the in-depth cells, no 2 clear distinct populations are revealed in flow cytometry. How do the authors explain this discrepancy?

Author Response

Dear Reviewer,

First of all, we would like to express our deep gratitude for careful evaluation of our manuscript and your competent comments. We have thoroughly revised the manuscript according to your remarks. Please find below the detailed answers to the questions mentioned in your review (Reviewer’s comments are presented in bold, our answers follow in plain text).

Comments and Suggestions for Authors:

Extensive editing of English language and style required

The English was carefully revised.

The manuscript entitled “Size-dependent penetration efficiency of antitumor agents in ovarian cancer spheroids” reports how the HER2-specific targeted toxin DARPin-LoPE poorly penetrates into three-dimensional spheroids.

General comment: To me, the unique cell line used in this article shows a very epithetial phenotype likely making spheroids highly dense. It would have been interesting to look at whether more invasive cells like the SKOV-3 derivative, SKOV3.ip1, make the same kind of spheroids and whether DARPin-LoPE and/or BSA might show a better penetration in such spheroids. This is important since others (Winner et al.) have shown that high molecular weight antibodies might enter into this 3D cell model. More generally, it would have been interesting to compare different cell lines with different invasive phenotypes since chemotherapeutics are mostly used to treat advanced diseases.

We thank the Reviewer for the comment. Along with the SKOV-3 and SKOV-kat cell lines, we also obtained spheroids from the SKOV3.ip1 cell line (please, see Figure R1 in the attached file). The cells of this line also developed well-formed spheroids with a clearly defined border, that is, they did not morphologically differ from the SKOV-3 (SKOV-kat) spheroids. For this study, we chose the SKOV-3 cell line (and its derivative SKOV-kat) to continue our previous work with this tumor model in studying the therapeutic potential of HER2-specific targeted toxin (Zdobnova et al., Oncotarget, 2015, doi: 10.18632/oncotarget.5130).

With regard to the work by Winner and colleagues, we see no critical contradictions in the results obtained. For the HER2-specific antibody Pertuzumab, Winner and colleagues showed 50 μm penetration into spheroids after 12-h incubation (which coincides precisely with our data for high-molecular agent DARPin-LoPE with the same incubation time, please see Fig. 2 of the manuscript). The deeper Pertuzumab penetration after 24-h incubation (about 150 μm, compared to ~70 μm DARPin-LoPE penetration in our experiments) can be explained by the looser structure of the SKOV3.ip1-RFP spheroids shown in Fig. 2 in Winner et al. The latter, in turn, may be due to different conditions for obtaining spheroids (two different ultra-low adhesive surfaces were used in our experiments and in Winner et al. work), as well as due to potential effect of RFP gene transfection on the adhesive properties of cells (this fact was observed in our laboratory when producing SKOV-3 and SKOV3.ip1 tumor cell transfectants with fluorescent protein genes).

We thank the Reviewer for the valuable suggestion to use a number of cell lines that differ in invasive potential, and we will definitely address it in our subsequent studies.

Other comments are detailed below:

Comment 1:  In the introduction section, the authors state that “However, when testing such agents in vivo, a significant decrease in their effectiveness is revealed. If such agents further undergo clinical trials no superiority is often observed compared to conventional treatment”. Such assumptions should be carefully used since anti-Her2 therapeutic strategies have shown activities in clinic and demonstrated superiority compared to conventional intervention.

We corrected the Introduction section according to the Reviewer remark.

Comment 2: On Figure 6, the authors disaggregated DARPin-LoPE treated 3D spheroids and analyzed cells by flow cytometry. While in microscopy, the authors show a clear difference between the labeling of the cells located at the surface of the spheroids compared to the in-depth cells, no 2 clear distinct populations are revealed in flow cytometry. How do the authors explain this discrepancy?

We believe that the number of molecules that have penetrated from layer to layer into the depth of the spheroid decreases along the gradient. In this regard, we did not expect to see division into 2 distinct populations of unstained and stained cells in FACS histograms. In the case of DARPin-LoPE, which penetrated poorly and stained only a small portion of cells in the spheroid, we observe a small right tail in 4- and 7-h histograms, corresponding to the cells with a higher signal level. For longer times, the entire population shifts somewhat toward a larger signal, while also preserving the tail on the right.

In the case of doxorubicin, which penetrated well and stained a significant portion of cells in the spheroid, we observe a shift of the main population towards an increase in the signal, but all the curves also show a left tail (reflecting, presumably, weakly stained cells deep in the spheroid).

Reviewer 3 Report

Targeted therapies for the treatment of cancer with large protein-based pharmaceutically active agents become more and more successful; however, the efficiency of delivering such agents into a tumor largely determines the prospects for its clinical use. Penetration of such large molecules into the inner areas of tumor tissue is critical.  In their study, the authors investigate the penetration of doxorubicin (0.5 kDa) and a protein-based targeted toxin (42 kDa) into the spheroids of human HER2-overexpressing ovarian adenocarcinoma.  They showed that the low penetration of the targeted toxin into spheroid correlates with a significant decrease in its efficiency against the three-dimensional model of the tumor spheroid as compared with the two-dimensional monolayer culture.  All experiments are conclusive and suitable controls were conducted.  The results are appropriately discussed, and important literature was taken into consideration.  The authors provide a valuable method, in particular as they avoid animal experiments by using spheroid models, to investigate tumor penetration of small and large-size drugs, which can be valuable for a broad readership.  They correctly mention that the values of the penetration degree of various agents differ significantly and that such differences may be caused by a variety of factors such as biological issues (cell-cell junctions density, degree of vascularization, stiffness of extracellular matrix, etc.) and physicochemical properties (size, surface charge, surface functionalization).  As this is the case, it is disputable whether the investigation of two substances is sufficient to correlate size and penetration depth.

Specific comments:

1.     The authors investigate doxorubicin (0.5 kDa) as an example for a small drug and the  protein-based targeted toxin DARPin-LoPE (42 kDa) as a large drug.  The results are convincing, but the interpretation that size correlates with the penetration efficacy is critical.  Of course, it is true for these two substances, but the question whether size is causal for the effect is not solved here.  Other properties of DARPin-LoPE and also of the target receptor might play a role (some are discussed by the authors) for weak penetration.  As the authors used doxorubicin as a small molecule, effects caused by HER2 cannot be reflected here.  Therefore, the manuscript and its conclusion would be more significant if a third substance would be included and ideally, all three substances should interact with the same receptor.  For instance, the use of lapatinib (0.581 kDa), DARPin-LoPE (42 kDa) and trastuzumab (145.5 kDa) or pertuzumab would represent a triplet of active substances of different size all interacting with HER2.  This would provide stronger evidence for a size-dependent penetration than the drug pair used by the authors.  Since “size-dependent penetration” is the major conclusion as indicated in the manuscript title, this should be taken into consideration.  In the current form, evidence is rather provided for “substance-dependent penetration” than for “size-dependent penetration”.

2.     Fig. 7A, B:  These results would be better to interpret if the same concentrations of the drug would be shown for the spheroid volume and the viability.

3.     Line 63: Should not be “various molecular weight” but “two distinct molecular masses”.

4.     DARPin should be briefly explained for readers not familiar with this designed protein class.

5.     Line 302: Sentence ended with “and”.  Part after “and” is missing.

6.     Line 361: What does “(link)” mean here?

Author Response

Dear Reviewer,

We would like to express our sincere gratitude for suggested improvements and valuable advice.

We have followed your recommendations and corrected the manuscript. Please find below the detailed answers to the questions mentioned in your review (Reviewer’s comments are presented in bold, our answers follow in plain text).

Comments and Suggestions for Authors:

English language and style are fine/minor spell check required

The English was carefully revised.

Targeted therapies for the treatment of cancer with large protein-based pharmaceutically active agents become more and more successful; however, the efficiency of delivering such agents into a tumor largely determines the prospects for its clinical use. Penetration of such large molecules into the inner areas of tumor tissue is critical.  In their study, the authors investigate the penetration of doxorubicin (0.5 kDa) and a protein-based targeted toxin (42 kDa) into the spheroids of human HER2-overexpressing ovarian adenocarcinoma.  They showed that the low penetration of the targeted toxin into spheroid correlates with a significant decrease in its efficiency against the three-dimensional model of the tumor spheroid as compared with the two-dimensional monolayer culture.  All experiments are conclusive and suitable controls were conducted.  The results are appropriately discussed, and important literature was taken into consideration.  The authors provide a valuable method, in particular as they avoid animal experiments by using spheroid models, to investigate tumor penetration of small and large-size drugs, which can be valuable for a broad readership.  They correctly mention that the values of the penetration degree of various agents differ significantly and that such differences may be caused by a variety of factors such as biological issues (cell-cell junctions density, degree of vascularization, stiffness of extracellular matrix, etc.) and physicochemical properties (size, surface charge, surface functionalization).  As this is the case, it is disputable whether the investigation of two substances is sufficient to correlate size and penetration depth.

 Specific comments:

1.     The authors investigate doxorubicin (0.5 kDa) as an example for a small drug and the  protein-based targeted toxin DARPin-LoPE (42 kDa) as a large drug.  The results are convincing, but the interpretation that size correlates with the penetration efficacy is critical. Of course, it is true for these two substances, but the question whether size is causal for the effect is not solved here. Other properties of DARPin-LoPE and also of the target receptor might play a role (some are discussed by the authors) for weak penetration.  As the authors used doxorubicin as a small molecule, effects caused by HER2 cannot be reflected here. Therefore, the manuscript and its conclusion would be more significant if a third substance would be included and ideally, all three substances should interact with the same receptor.  For instance, the use of lapatinib (0.581 kDa), DARPin-LoPE (42 kDa) and trastuzumab (145.5 kDa) or pertuzumab would represent a triplet of active substances of different size all interacting with HER2.  This would provide stronger evidence for a size-dependent penetration than the drug pair used by the authors.  Since “size-dependent penetration” is the major conclusion as indicated in the manuscript title, this should be taken into consideration.  In the current form, evidence is rather provided for “substance-dependent penetration” than for “size-dependent penetration”.

In this article, we discuss and experimentally confirm the hypothesis about the importance of the molecule size as one of the factors determining the effectiveness of the penetration of the agent into the tissue (in our case, into the tumor spheroid model). We completely agree with the Reviewer's comment that the study of two agents is not enough to unambiguously establish a correlation between the size of the molecule and the depth of its penetration. We changed the title of the article in order to reflect the obtained results more correctly.

We thank the Reviewer for the proposed options for HER2-specific agents, this approach is of undoubted interest for our future research. When designing our work, the priority was to study living spheroids by minimally invasive methods in order to minimize the influence of sample preparation procedure on the results and their interpretation. In this regard, to visualize the penetration of agents into the spheroid, we have chosen the confocal microscopy of living spheroids. In line with this, agents of different molecular weights were selected based on their own fluorescence (doxorubicin) or the ability to label them with fluorescent dye with minimal impact on the molecular weight and functionality (amino-specific labeling of DARPin-LoPE and BSA proteins with low-molecular-weight dyes). Fluorescent properties of lapatinib are also shown (PMID: 25820099), however, their strong dependence on receptor binding and aggregation was shown, which would make it difficult to interpret data on its penetration based on the detection of its fluorescence.

To evaluate the effect of the interaction of the targeted toxin with HER2, we compared its penetration with that of BSA – protein of the similar molecular mass but with no HER2 specificity. The penetration depth of these two agents did not differ (please, see Fig. 2 and 4 in the manuscript) that confirms the conclusion about the possible role of molecule size.

We emphasize that we consider size factor as one of the most important, but not the only, among those influencing drug penetration.

2.     Fig. 7A, B:  These results would be better to interpret if the same concentrations of the drug would be shown for the spheroid volume and the viability.

Figure 7 and its caption were modified.

3.     Line 63: Should not be “various molecular weight” but “two distinct molecular masses”.

Corrected (lines 63-74).

4.     DARPin should be briefly explained for readers not familiar with this designed protein class.

The appropriate details were added (lines 227-229).

5.     Line 302: Sentence ended with “and”.  Part after “and” is missing.

Corrected.

6.     Line 361: What does “(link)” mean here?

Corrected.

Round 2

Reviewer 1 Report

All comments addressed, however, I recommend to include a discussion of the apoptosis stainings performed.

Author Response

Dear Reviewer,

We would like to express our deep gratitude for evaluation of our manuscript and your competent comments. Please find below the detailed description of the revisions (Reviewer’s comments are presented in bold, our answers follow in plain text).

All comments addressed, however, I recommend to include a discussion of the apoptosis stainings performed.

The appropriate text was added to the manuscript (please, see lines 323-326). The details of the experiment were included in the Supplementary materials (Fig. S3).

Reviewer 2 Report

The authors took into account all my comments and responded to them adequately. In a whole, the manuscript has been significantly improved.

Author Response

Dear Reviewer,

We would like to express our sincere gratitude for evaluation of our manuscript.

Comments and Suggestions for Authors:

The authors took into account all my comments and responded to them adequately. In a whole, the manuscript has been significantly improved.